# Prohibitin Links Cell Cycle, Motility and Invasion in Prostate Cancer Cells

**DOI:** 10.3390/ijms24129919

**Published:** 2023-06-08

**Authors:** Sarah Koushyar, Pinar Uysal-Onganer, Wen Guo Jiang, Dafydd Alwyn Dart

**Affiliations:** 1Cardiff China Medical Research Collaborative, School of Medicine, Cardiff University, Cardiff CF14 4YS, UK; 2Cancer Mechanisms and Biomarkers Research Group, School of Life Sciences, College of Liberal Arts and Sciences, University of Westminster, 115 New Cavendish Street, London W1W 6UW, UK; 3Institute of Medical and Biomedical Education, St George’s University of London, Cranmer Terrace, Tooting, London SW17 0RE, UK

**Keywords:** prostate cancer, WNT signalling, PHB

## Abstract

Prohibitin (*PHB*) is a tumour suppressor gene with several different molecular activities. PHB overexpression leads to G1/S-phase cell cycle arrest, and PHB represses the androgen receptor (AR) in prostate cancer cells. PHB interacts with and represses members of the E2F family in a manner that may also be AR-linked, therefore making the AR:PHB:E2F interaction axis highly complex. *PHB* siRNA increased the growth and metastatic potential of LNCaP mouse xenografts in vivo. Conversely, *PHB* ectopic cDNA overexpression affected several hundred genes in LNCaP cells. Furthermore, gene ontology analysis showed that in addition to cell cycle regulation, several members of the *WNT* family were significantly downregulated (*WNT7B*, *WNT9A* and *WNT10B*), as well as pathways for cell adhesion. Online GEO data studies showed *PHB* expression to be decreased in clinical cases of metastatic prostate cancer, and to be correlated with higher WNT expression in metastasis. *PHB* overexpression reduced prostate cancer cell migration and motility in wound-healing assays, reduced cell invasion through a Matrigel layer and reduced cellular attachment. In LNCaP cells, *WNT7B*, *WNT9A* and *WNT10B* expression were also upregulated by androgen treatment and downregulated by androgen antagonism, indicating a role for AR in the control of these *WNT* genes. However, these *WNTs* were strongly cell cycle regulated. *E2F1* cDNA ectopic expression and *PHB* siRNA (both cell cycle promoting effects) increased *WNT7B*, *WNT9A* and *WNT10B* expression, and these genes were also upregulated as cells were released from G1 to S phase synchronisation, indicating further cell cycle regulation. Therefore, the repressive effects of PHB may inhibit *AR*, *E2F* and *WNT* expression and its loss may increase metastatic potential in human prostate cancer.

## 1. Introduction

Prostate cancer (PC) is the most common cancer amongst men in the UK, with approximately 52,000 cases being diagnosed each year (Cancer Research UK, UK, 2016–2018, ICD-10 C61). If left undetected, PC often metastasizes to sites such as the bone and the lymph nodes. The main pathogenic pathway associated with the progression of PC is the androgen receptor (AR) signalling cascade [1,2]. Aberrant activation of AR signalling results in the heightened transcription of genes associated with cell cycle progression and cellular metastasis.

Although there has been a steady decline in the cases of newly diagnosed metastatic PC recently, the risk of clinically localized PC transitioning to extra-prostatic disease is still high (30–60%). The initial stages of PC metastasis begin with the cells adopting a mesenchymal state, allowing local migration to the pelvic lymph nodes; here, the migratory cells can enter the local vasculature and, if successful in surviving the harsh environment of the bloodstream, can disseminate to distant organs such as the bones [3,4].

Often in PC, the AR-associated co-factors can themselves become dysregulated and can influence the activity of the AR [5]. One well-established co-repressor of the AR is prohibitin (PHB), which was discovered in the 1980s as possessing tumour suppressor functions [6]. It is an ubiquitously expressed protein, mainly residing within the inner membrane of the mitochondria where it interacts with its homologue PHB2, maintaining the stability and structure of the mitochondria [7]. However, it has also been shown to localize within the nucleus, where it interacts with the E2F transcription factor gene family [8,9]. E2Fs are heavily implicated in the regulation and the progression of the cell cycle. Therefore, PHB, similar to the function of RB1, is able to reduce E2F functions, thus inhibiting the cell cycle. This phenomenon has been identified in both androgen-dependent PC and oestrogen-dependent breast cancer [10,11,12,13].

The interaction of AR and PHB is very complex. Previously, we have shown that *PHB* is downregulated by androgen treatment at the gene and protein level via transcriptional mechanisms and it can be downregulated post-transcriptionally via the androgen regulated miR-27a [14]; more recently, we have shown that the non-genomic actions of androgens, via cellular signalling cascades, can induce a rapid dephosphorylation of PHB, reducing its interaction with E2F [9].

The resultant lowering of PHB functional activity can increase AR-mediated PC growth. Indeed, *PHB* siRNA increased PC cell and xenograft growth, and interestingly, increased the sensitivity of the AR towards weaker adrenal androgens. Conversely, ectopic overexpression of *PHB* cDNA led to cell cycle arrest at G1/S, and a reduction in prostate cell and xenograft growth [11,12,13].

Here, we have reported that the AR-regulated protein, PHB, affects and influences the metastatic potential and migration of prostate cancer cells through a cell-cycle-linked pathway involving several members of the WNT family of proteins.

## 2. Results

### 2.1. Doxycycline-Induced PHB siRNA Increases Tumour Growth and Increases Metastatic Potential

In previously published studies [9,12,13], we used the hormone-dependent LNCaP cell line, stably transfected with an integrated doxycycline-inducible system for either ectopic *PHB* cDNA expression or *PHB* siRNA. Ectopic expression of PHB resulted in rapid cell cycle arrest at the G1/S boundary, with the downregulation of several genes involved in the G1 to S phase transition [9]. RNA-seq analysis indicated that the *E2F* family was the main pathway affected, followed closely by the *WNT* signal transduction and cell adhesion pathways. Additionally, with *PHB* cDNA, a strong reduction in LNCaP xenograft growth and AR transcriptional activity was observed.

Conversely, *PHB* siRNA resulted in increased cell cycle entry, increased cell and xenograft growth, and increased AR activity. When LNCaP/Luc/PHB^siRNA^ xenografts were grown in both flanks of nude mice, we noted an increase in tumour growth with doxycycline-induced *PHB* siRNA (summarised in Figure 1A), as previously published by Dart et al., 2009 [12]. Interestingly, as the LNCaP cells used contained an integrated AR luciferase reporter, in 3/10 mice we detected an additional luciferase signal emanating from outside the tumour—indicative of likely metastatic deposits (Figure 1B). Upon ex vivo tissue examination, unusual growths were seen in the liver and kidneys of the mice, which showed luciferase activity. Upon histological examination of liver sections, large deposits of AR^+ve^ cells were detected using an anti-human AR antibody, showing cells with large nuclei that were not present in a normal mouse liver nor in mice with LNCaP/Luc/PHB^siRNA^ xenografts without doxycycline treatment (Figure 1C,D). This was indicative that the original LNCaP/Luc/PHB^siRNA^ cells had spread to produce metastatic deposits beyond the original xenograft.

### 2.2. PHB Expression Is Reduced in Advanced Metastatic PC

Although LNCaP cells are derived from lymph node metastasis in humans, their capacity to form metastasis from solid xenografts in nude mice is relatively low; however, it has been reported for more aggressive sublines [15,16]. Therefore, we hypothesised that reduced PHB was associated with an increased capacity for metastasis. To analyse if this was true for human prostate cancers, we investigated online GEO clinical datasets to assess if *PHB* downregulation was associated with metastatic PC. Analysis of GEO dataset GSE6919 [17], comprising of prostate cancer tissue samples, showed that *PHB* gene expression was significantly downregulated as the cancer progressed from the primary tumour to a metastatic state (Figure 2A). Similarly, in GEO dataset GDS1439 [18], there was a significant reduction in *PHB* gene expression from benign prostate tissue to metastatic PC (Figure 2B). We evaluated the *WNT* family expression in the same datasets (GSE6919 and GDS1439). *WNT7B*, *WNT9A* and *WNT10B* were all increased in metastatic prostate cancer, compared to the normal prostate (Figure 2A,B). For *WNT7B* and *WNT9A*, *WNT* expression showed an inverse correlation with *PHB* levels. *WNT10B* showed an inverse correlation in one dataset only. *AR* and *PHB* also showed an inverse correlation in these studies, as expected (Appendix A).

Interestingly, another dataset (GDS3973) [19] showed that *PHB* gene expression was downregulated in two docetaxel-resistant prostate cancer cell lines (Appendix A). *PHB* was also downregulated in an invasive gastric adenocarcinoma (dataset GDS4198 [20] when compared to proliferative gastric adenocarcinoma, suggesting *PHB*’s role in cancer progression is not PC-specific (Appendix A).

Utilising the Gepia website (http://gepia.cancer-pku.cn/about.html, accessed on 1 February 2023), we examined the disease-free survival of the prostate cancer dataset (PRAD). Although, no significant difference was seen for *PHB*, higher expression of all *WNTs* was associated with a reduced disease-free interval, which was significant for *WNT7B* and *WNT10B*, *p* > 0.01 (Appendix A).

### 2.3. PHB Manipulation Affects WNT Gene Expression

The WNT signal transduction pathway was strongly downregulated by *PHB* ectopic expression, as determined previously by RNA-seq [9], shown in Figure 3A, as analysed using the IPA (Ingenuity Pathway Analysis, version 2.4) software. *PHB* cDNA ectopic expression resulted in protein levels that were approximately seven times higher than that seen in normal LNCaP cells [9,12]. Several gene members of the *WNT* family were downregulated, in particular *WNT7B*, *WNT9A* and *WNT10B*, in response to *PHB* overexpression (*WNT7B p* < 0.001, *WNT9A p* < 0.01, *WNT10B p* < 0.05; *n* = 3, see Figure 3B). We then further validated this downregulation by qPCR from doxycycline-treated LNCaP/Luc/PHB^cDNA^ cells (Figure 3C). Conversely, doxycycline-induced PHB^siRNA^ increased *WNT7B*, *WNT9A* and *WNT10B* in these cells (Figure 3D).

### 2.4. PHB Overexpression Inhibits the Migration and Adhesion of Both Androgen-Dependent and Androgen-Independent PC Cells

Given that the WNT family of proteins are strongly implicated in epithelial–mesenchymal transition (EMT) and in the metastatic process, coupled with an increased metastatic potential seen with *PHB^siRNA^* in LNCaP xenografts, we set out to evaluate the relationship between PHB and these WNT family members. Aberrant activation of WNT signalling is implicated in tumorigenesis. The WNT protein family is also implicated in the key migration process known as the epithelial–mesenchymal transition (EMT), essential for cancer cell metastasis.

Therefore, the effects of PHB overexpression, and the associated downregulated WNT genes on prostate cell migration was analysed via several functional assays. Additionally, we carried out assays on both androgen-dependent (LNCaP) and androgen-independent (PC3) prostate cell lines, to assess PHB’s role on PC cell motility. The highly metastatic PC3 prostate cancer cell line has previously been shown to express low PHB levels [9]. Assays used were the transwell filter migration assay and a wound (scratch) healing assay. In LNCaP/PHB^cDNA^, *PHB* was overexpressed ectopically via doxycycline treatment (seven-fold approx.), and in PC3 cells, *PHB* was overexpressed via transient transfection of the pSG5-PHB plasmid (three-fold approx.). No doxycycline-inducible PC3 were available for this study.

The LNCaP/PHB^cDNA^ cell line treated with doxycycline showed that at 24, 32 and 48 h, there was a significant decrease in the migration of cells to seal the scratch/wound when compared to untreated cells (*n* = 3, *p* < 0.05) (Figure 4A). In synergy with the scratch assay, the transwell filter assay showed a significant reduction in migration when compared to untreated cells (*n* = 3, *p* < 0.05). Parallel experiments were carried out with the AR null cell line PC3s, where cells were left to for up to 20 h (Figure 4B and Appendix A).

Metacore analysis depicted WNT signalling and integrin-mediated cell adhesion as being among the top networks affected by PHB; thus, how PHB influences the attachment of cells to a basement membrane/Matrigel layer was assessed. LNCaP/Luc/PHB^cDNA^ cells with or without doxycycline (10 μM, 16 h) or PC3 cells transfected with empty PHB-expressing plasmids were seeded onto Matrigel-coated 96-well plates. PHB overexpression significantly reduced the number of adhered cells to the Matrigel (a decrease of three-fold) (*n* = 3, * *p* < 0.0001) (Figure 4C–E). In synergy with this experiment, siRNA targeted knock-down of PHB induced the opposite effect: an increase in adhered cells to the basement membrane was seen (Figure 4F). This highlights PHB’s role in decreasing cell to basement membrane adhesion—a process vital for the initiation of cancer metastasis.

*WNT7B*, *WNT9A* and *WNT10B*, when transfected into LNCaP cells, caused a modest increase in cell growth, although this was not statistically significant (Figure 5A). Additionally, *WNT7B*, *WNT9A* and *WNT10B* overexpression did induce increased motility in LNCaP cells in a wound-healing assay (Figure 5B).

### 2.5. WNT7B, 9B and WNT10B Are AR- and E2F1-Responsive

The main effect of overexpressing PHB in PC cell lines is the promotion of cell cycle arrest at the G1/S boundary, in part via the binding and inhibition of the E2F transcription factor [9]. Additionally, activation of the AR leads to downregulation and dephosphorylation of PHB, leading to increased signalling through E2Fs [9,11].

Therefore, we investigated whether the genes *WNT7B*, *WNT9A* and *WNT10B* were either AR-regulated, E2F-responsive or were cell-cycle-regulated. Firstly, we analysed if putative AR and E2F binding sites were present in the promoters of these genes. Using the Alggen website (http://alggen.lsi.upc.es/promo, accessed on 2 February 2019) and online ChIP-seq AR studies in LNCaP cells (GSE28219), we found that each gene had an E2F binding element, although the % similarity was not as great as those seen in genes strongly regulated by E2F, e.g., the *MCM* gene family. *WNT7B* had an androgen response element within in the gene structure, but not at the promoter (Appendix A). No AREs were seen in the promoters of *WNT9A* or *WNT10B*.

We tested androgen regulation of these *WNT* genes. LNCaP cells were starved in charcoal-stripped serum for 72 h, after which they were treated with 0–10 nM testosterone. *PSA* (*KLK3*) expression was used as a positive control and showed a strong gene induction, as expected. All three *WNT* genes showed a weak-to-modest induction with testosterone treatment, which was dose-dependent (Figure 6A). *WNT7B* showed the strongest induction. Additionally, when LNCaP cells were treated with bicalutamide (0–50 μM) to inhibit AR action, we saw a modest inhibition in the expression of all three *WNT* genes (Figure 6A).

We then transfected LNCaP cells with an *E2F1* expression plasmid and observed the E2F1-mediated upregulation of *WNT7B*, *WNT9A* and *WNT10B*, which was not observed with the empty vector control (Figure 6B).

### 2.6. WNT7B, WNT9A and 10B Expression Are Linked to the Cell Cycle

E2F1, PHB and the AR have a very complex pathway in determining cell cycle progression. E2F1 is the main key transcription factor in progressing the cell cycle into the S phase. PHB has been shown to inhibit the E2F family of proteins, and AR can inhibit *PHB* expression; the AR itself is a proposed licensing factor for cell cycle entry in some cells [21,22]. Therefore, it is very difficult to separate these components experimentally.

To study the expression of these genes during the cell cycle, we utilised isoleucine starvation to synchronise the cells. LNCaP and PC3 cells were synchronized in isoleucine-free media, and then released into normal media. LNCaP cells were kept in the presence of physiological levels of androgen, and AR-mediated effects were kept constant throughout. Cells were collected for FACS and RNA extraction at intervals as they re-entered the cell cycle.

When starved of isoleucine, the population of LNCaP cells in the S phase dropped significantly, from 16% to 3% (Figure 7A,B). When cells were transferred back to full media, the number of cells in the S phase increased, starting at 24 h and then increasing back to pre-starvation levels at 30–72 h. As the cells re-entered the S phase at 24 h there was a peak in *WNT7B*, *WNT9A* and *WNT10B* gene expression levels, which remained high until 30 h, and then fell back to previous levels (Figure 7C). Similarly, there was an increase in G1/S phase gene expression, e.g., *TYMS* (Thymidylate Synthase), *MCM4* & *MCM5*, and *cyclin D* (*CCND1*) expression (Figure 7C).

When starved of isoleucine, the population of PC3 cells in the S phase dropped significantly, from 25% to 7% (Figure 7D). When cells were transferred back to full media, the number of cells in the S phase increased from 6–24 h. As the cells re-entered the S phase, peaking at 24 h, there was a strong increase in *WNT7B*, *WNT9A* and *WNT10B* gene expression levels. Similarly, there was a strong increase in *TYMS*, *MCM4* & *MCM5*, and *CCND1* expression (Figure 7E).

As LNCaP cells were difficult to synchronise, we carried out a similar experiment using HeLa cells, which had been synchronised at G1/S following a thymidine/aphidicolin block. LNCaP cells did not survive this procedure. This allowed for a much tighter pool of cells moving through the S phase after release. As the cells moved into the S phase (Appendix A), an increased expression of both *WNT7B*, *WNT9A* and *WNT10B* was seen in the HeLa cells (Appendix A).

## 3. Discussion

We set out to further understand the complex role of PHB in the progression of PC. Previously, ectopic overexpression of *PHB* cDNA reduced prostate cell and xenograft growth; conversely, *PHB* siRNA increased cell and xenograft growth, and increased the androgen sensitivity of LNCaP cells [11,12,13]. *PHB* siRNA was seen to increase the metastatic potential of LNCaP/Luc/PHB^siRNA^ cells in vivo, causing metastatic deposits to form in the liver and kidneys, as visualised by luciferase expression. LNCaP cells do not readily form metastases, even with a high xenograft tumour burden, but they have been reported in sublines. Liver metastasis is rarer in prostate cancer, but has been recorded in several clinical datasets, as seen in those used in Figure 2.

In a previous RNA-seq study on ectopic *PHB* cDNA overexpression, we saw that the cell cycle and *WNT* signalling were the most affected pathways by far. We had already examined the cell cycle effects of PHB in detail in a previous study; however, here, we examined if the increased metastatic potential of LNCaP cells with decreased PHB was as a result of its effects on *WNT* expression.

WNTs have critical functions during normal embryogenesis and tissue homeostasis, regulating cell motility, adhesion, invasion, tissue patterning and proliferation—reviewed by the authors in [23]. WNT proteins bind to the Frizzled (FZD) receptors at the plasma membrane, initiating canonical or noncanonical intracellular signalling, leading to a highly complex array of cellular responses. Canonical WNT signalling is activated upon canonical WNT ligands binding to the FZD receptor, initiating the cell membrane recruitment of dishevelled (DVL) and AXIN. This causes a dissociation of the destruction complex, leading to an accumulation of cytoplasmic β-catenin. This can translocate to the nucleus and associate with members of the T-cell factor/lymphoid enhancer factor (TCF/LEF) to form a transcriptional activation complex, ultimately leading to the regulation of WNT gene expression. The noncanonical pathway is independent of β-catenin and can be divided into three pathways known as the planar cell polarity (PCP) pathway, the WNT/Ca^2+^ and the WNT/STOP pathway. The PCP pathway is activated via noncanonical WNT proteins (e.g., WNT5A), activating small GTPases, causing downstream activation of the JNK, ROCK and ATF2 signalling pathways. Activation of WNT/Ca^2+^ mediates protein kinase C (PKC), triggering cytoskeletal rearrangements and target gene transcription via either the nuclear factor of activated T cells (NFAT) or nuclear factor kappa B (NF-κB). The WNT/STOP pathway remains to be fully elucidated; however, it involves the stabilisation of proteins, regulating cell division through the LRP6-DVL signalling pathway.

WNTs have been strongly implicated in the invasiveness and metastasis processes in many cancer types. Changes in cell motility and morphology are essential during embryogenesis but are also critical factors in cancer cell motility and invasiveness—reviewed by the authors in [23,24,25]. WNT7B, WNT9A and WNT10B did indeed cause a small increase in LNCaP cell growth and motility, and WNT7B has been shown to be required for the growth of both androgen-dependent and castration-resistant prostate cancer cells in other studies [26].

Manipulating PHB in this study showed that some of the *WNT* family of genes were strongly affected. *PHB* overexpression strongly downregulated *WNT7B*, *WNT9A* and *WNT10B* expression in LNCaP cells. *PHB* ectopic cDNA expression inhibited cell migration and adhesion, with converse effects seen with *PHB* siRNA. *WNT7B* has been strongly implicated in PC, being a direct AR target gene highly expressed in castration-resistant prostate cancer (CRPC) cells [26]. *WNT7B* and *WNT9A* have been significantly associated with the biochemical recurrence of PC [27]. *WNT10B* has a more complex association, being highly expressed in the neonatal prostate, low in adults, and then high in metastatic cancers and in cell lines with a high metastatic potential [28].

The expression of *WNT7B*, *WNT9A* and *WNT10B* were modestly androgen-inducible, and have been previously described [26]. Since PHB is a strong repressor of the AR, it is likely that the observed *PHB*-cDNA-mediated repression of the *WNT* genes could be mediated via the PHB:AR axis. Indeed, AR inhibition by the androgen antagonist, bicalutamide, reduced expression by over 50%.

However, the increased expression of *WNT7B*, *WNT9A* and *WNT10B* were also seen in LNCaP cells, when they entered the cell cycle from a synchronised (isoleucine starved) population, in the presence of constant levels of testosterone. The expression of *TYMS*, *MCM5* and *CCND1* also confirmed the G1/S-phase specificity of these cells. Similarly, the non-AR-dependent cell line PC3 and the non-prostate cell line HeLa both also strongly expressed these *WNT* genes when they entered the cell cycle—progressing from the G1 to S phase, indicating that AR may not be the only controlling factor. PC3 cells have a much lower level of endogenous PHB and have a much greater migratory and invasive phenotype in vitro, and metastatic potential in vivo.

PHB also represses the E2F transcription factor family, and the transfection of E2F1-expressing plasmids increased the expression of these *WNT* genes in LNCaP cells, corroborating the cell-cycle-controlled nature of these genes. Similar results were seen when PHB was reduced by siRNA. These data therefore show a very complex pattern of expression for these *WNT* genes. This is additionally compounded by the complex association and interplay between PHB, E2F1 and AR in licensing and coordinating the cell cycle.

Given the clinical data of reduced *PHB* levels between primary and metastatic tumours, and the increased growth and metastatic potential seen in LNCaP xenografts with reduced *PHB* levels, we hypothesised that PHB does indeed have a strong role in PC progression. As PC cells progress from the primary tumour to cells with metastatic potential, *PHB* levels are reduced. *PHB* levels may be reduced by increased activity of the AR, which downregulates *PHB* via various mechanisms, e.g., microRNA27a-mediated targeting of the *PHB* UTR [14], promoter downregulation, and dephosphorylation of the protein itself [9]. This, in turn, reduces the repression of the AR and invokes a positive feedback cycle on AR activity, which can increase AR sensitivity to low androgen levels and other weaker adrenal androgens [13]. Additionally, reduced PHB reduces the repressive effects on E2F1, enabling increased cell cycle entry and increased expression of cell-cycle-regulated genes, e.g., *MCM4* & *MCM5s*. WNT expression may be similarly increased. Increased WNT expression then increases the migration and invasion properties of prostate cancer cells. Indeed, in the GEO datasets presented here (GSE6919 and GDS1439), *WNT7B*, *WNT9A* and *WNT10B* were all increased in metastatic prostate cancer, where *PHB* levels were reduced.

## 4. Materials and Methods

### 4.1. Cell Line Maintenance

Cells were maintained at 37 °C in 5% CO_2_. LNCaP cells were maintained in RPMI medium with 10% foetal bovine serum (First Link, Wolverhampton, UK) (ATCC CRL-1740). LNCaP/Luc/PHB^cDNA^ and LNCaP/Luc/PHB^siRNA^ cells were maintained in RPMI medium with 10% doxycycline-free foetal bovine serum (Clontech, Palo Alto, CA, USA), 12 μg/mL blasticidin (Sigma, Dorset, UK), 0.3 mg/mL Zeocin (ThermoFisher, Waltham, MA, USA) and 500 μg/mL G418 (Sigma). HeLa (ATCC CCL-2) and PC3 (ATCC CRL-1435) were maintained in Dulbecco’s modified Eagle’s medium with 10% foetal bovine serum. Media were supplemented with 2 mM L-glutamine, 100 U/mL penicillin and 100 mg/mL streptomycin (Sigma). Seventy-two hours prior to androgen exposure, the medium was replaced with the ‘starvation medium’ consisting of phenol-red-free RPMI medium (or Dulbecco’s modified Eagle’s medium), supplemented with 5% charcoal-stripped foetal bovine serum (First Link).

### 4.2. Transfection and Plasmids

LNCaP and PC3 cells were transiently transfected using Lipofectamine 2000 (ThermoFisher), following the manufacturer’s protocols. The pcDNA empty vector (Invitrogen), pSG5-PHB, was previously produced (Ref: Gamble 2004, Oncogene). WNT7B, WNT9A and WNT10B plasmids were obtained from Addgene (pcDNA-WNT7B, pc-DNA-WNT9A, pc-DNA-WNT10B was a gift from Marian Waterman (Addgene plasmids # 35915, 35921, 35918; http://n2t.net/addgene, accessed on 2 February 2018).

### 4.3. Cell Synchronization

#### 4.3.1. Isoleucine Synchronization

Isoleucine-deprivation was carried out by first allowing the cells to become sub-confluent and then replacing the media with isoleucine-free media (ThermoFisher) supplemented with 6% dialyzed fetal calf serum (FCS) (ThermoFisher), 2.5 mM glutamine, 100 mg/mL streptomycin and 100 U/mL penicillin. A total of 10 nM testosterone was added to LNCaP cells. Cells were maintained in isoleucine-free media for 36–40 h. Cells were then released from the isoleucine block by replacing the media with complete media containing 10% FCS.

#### 4.3.2. HeLa Cell Synchronization

Synchronization of cells at the G1/S phase was achieved by incubation of HeLa cells for 25 h in 2.5 mM thymidine (Sigma). Cells were subsequently released from the thymidine block and allowed to grow for a further 12 h in the culture medium prior to incubation in 5 μg/mL aphidicolin (Sigma) for 20–24 h. Cells were washed and the media was replaced with complete media. Synchronizing cells at the M phase was achieved by incubation of HeLa cells for 25 h in 2.5 mM thymidine, followed by washing and releasing cells into a normal medium for 6 h, followed by subsequent incubation in 50 ng/mL nocodazole for 12 h. Cells were then washed again and released from the M phase into a normal medium.

### 4.4. Wound-Healing/Scratch Assay

Cells were seeded onto a 24-well plate until a confluent monolayer was formed. A wound was made using a p10 tip. EVOS-FL auto was used to photograph and measure the wound closure over a 24 h period under the environmental conditions of 5% CO_2_ at 37 °C. Each condition had triplet wells, and within each well, three points of references were taken of the scratch. Semi-quantification of the wound area was measured using Image J software, version 1.51 (National Institutes of Health, Bethesda, MD, USA).

### 4.5. Transwell Filter Migration Assay

Cells (50,000) were seeded onto the upper chamber of a 8 μM ThinCert™ insert (Greiner-Bio One, Stonehouse, UK) in 500 μL of 1% (*v*/*v*) FCS RPMI. The lower chamber of the 24-well plate contained 1 mL of 10% FCS RPMI to create a chemoattractant. Cells were left to migrate for up to 24 h, at which time, filters were washed with PBS and placed onto a fresh 24-well plate. An enzyme-free cell dissociation solution (Millipore, Burlington, MA, USA) (350 μL) made with calcein AM (Thermo Fisher Scientific) (ratio of 1.2 μL:1 mL) was placed in the lower chamber of the 24-well plate and left to incubate at 37 °C for 1 h. The solution was transferred to a black 96-well plate and fluorescence was measured using the Glomax multi-detection system (Promega, Madison, WI, USA) at excitation and emission wavelengths of 495/515 nm, respectively.

### 4.6. Cell to Basement Membrane Adhesion Assay

A Matrigel™ basement membrane was coated onto wells of a 96-well plate (50 μg/mL) in serum-free media. Cells were seeded onto the Matrigel™ at a cell density of 50,000 (LNCaPs) and 20,000 (PC3s) and left to seed for 1.5 h. Cells were then fixed to the Matrigel™ with 4% formalin (*v*/*v*) and then stained with 0.5% crystal violet solution (*v*/*v*). Plates were gently washed with tap water and left to air-dry overnight. Cells were imaged using a light microscope.

### 4.7. RNA Extraction and Quantitative SYBR^®^ Green PCR (Q-PCR)

Total RNA samples were prepared following the manufacturer’s guidelines using the TRI reagent (Sigma-Aldrich). Following RNA extraction, RNA was reverse transcribed to cDNA following the manufacturer’s guidelines for GoScript™ Reversion Transcription Mixes (Promega).

Reactions were performed in triplicate on an ABI Prism 7900HT System (Applied Biosystems, Warrington, UK). Reactions consisted of 2 μL cDNA, 2 μL water, 5 μL 2× SYBR green PCR Master Mix (Applied Biosystems) and 1 μL specific primers (0.25 pmol/μL). Primer details are given in the Supplementary Information. The parameters used were as follows: 50 °C for 2 min, 95 °C for 10 min and 40 cycles of 95 °C for 15 s and 60 °C for 1 min. Levels were normalized to *GAPDH*, *RPL19* and *ACTB*.

## 5. Conclusions

PHB overexpression reduces the expression of several *WNT* family members in LNCaP prostate cancer cells and reduces motility and invasiveness. Conversely, a reduction in *PHB* is correlated with an increased metastatic potential, both in mouse xenograft models and in GEO clinical datasets. PHB represses AR and E2F, and links the expression of *WNT7B*, *WNT9A* and *WNT10B* to the cell cycle. Further exploration is needed to fine tune the exact molecular mechanisms underpinning this complex network and to determine whether PHB can be therapeutically exploited to stop PC progression from a primary to a metastatic state.

## Figures and Tables

**Figure 1 ijms-24-09919-f001:**
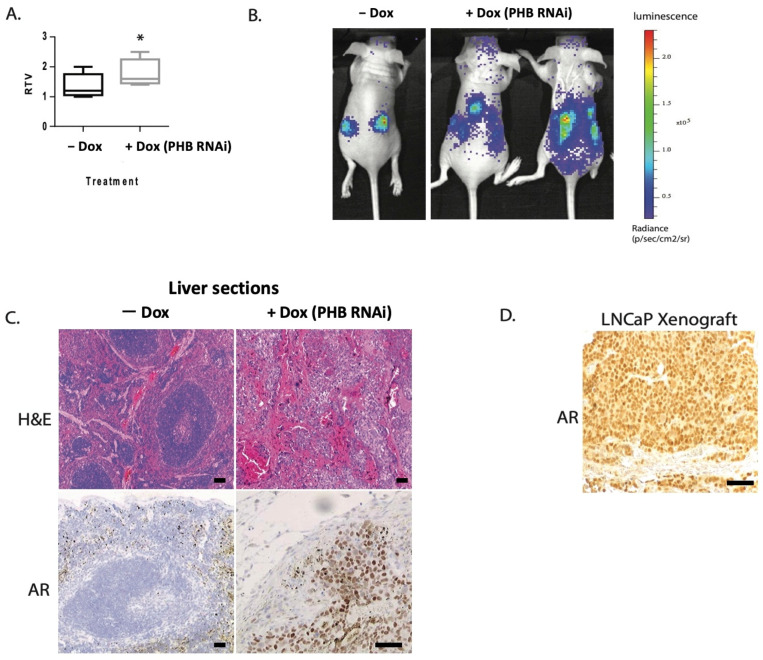
PHB siRNA increases tumour growth and increases metastatic potential. (**A**) Boxplot showing the relative tumour volume of LNCaP/Luc/PHB^siRNA^ xenografts grown in both flanks of nude mice with or without doxycycline (in drinking water) at 36 days. Data represent the mean + S.E.M. *n* = 8. (**B**) Bioluminescent imaging of luciferase activity in mice harbouring LNCaP/Luc/PHB^siRNA^ xenografts, treated with or without doxycycline. Luciferin substrate via intraperitoneal injection. (**C**) Immunohistochemical staining of liver tissue sections from mice harbouring LNCaP/Luc/PHB^siRNA^ xenografts, treated with or without doxycycline. Upper panel shows haematoxylin and eosin chemical staining. Lower panel shows immunostaining for AR. (**D**) Immunohistochemical staining of sections of LNCaP/Luc/PHB^siRNA^ xenografts for AR. * *p* ≤ 0.05. Scale bar = 100 μm.

**Figure 2 ijms-24-09919-f002:**
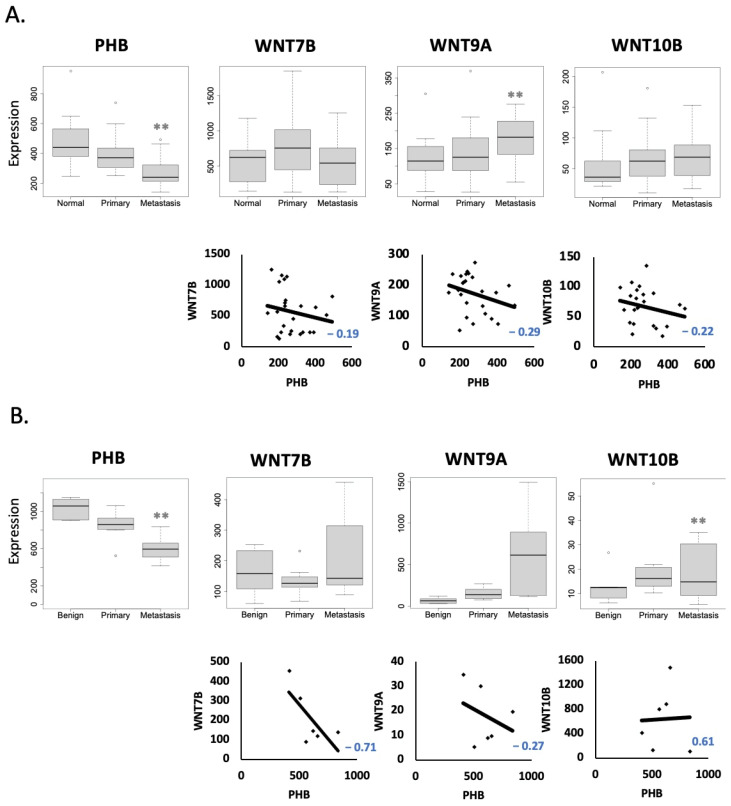
*PHB* expression in metastatic prostate cancer is inversely correlated with *WNT* family members *WNT7B*, *WNT9A* and *WNT10B*. (**A**) Boxplots showing *PHB* and *WNT7B*, *WNT9A* and *WNT10B* expression values from the GSE6919 dataset. Data are separated into normal prostate tissue (*n* = 17), primary prostate cancer (*n* = 66) and metastatic prostate cancer (*n* = 25). Graphs underneath display the regression analysis of *PHB* expression versus *WNTs* from metastasis samples. (**B**) Boxplots showing *PHB* and *WNT7B*, *WNT9A* and *WNT10B* expression values from the GSD1439 dataset. Data are separated into benign prostate tissue (*n* = 6), primary prostate cancer (*n* = 7) and metastatic prostate cancer (*n* = 6). Graphs underneath display the regression analysis of *PHB* expression versus WNTs from metastasis samples. ** *p* ≤ 0.01.

**Figure 3 ijms-24-09919-f003:**
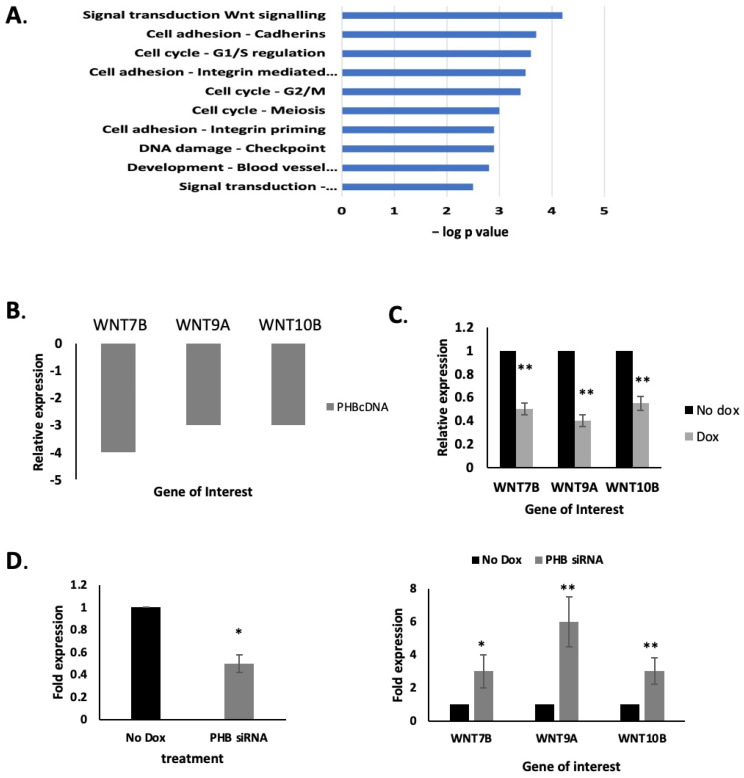
PHB manipulation regulates WNT gene expression. (**A**) Bar chart indicating the most significant gene networks influenced by PHB overexpression. Produced in Metacore software (https://clarivate.com, accessed on 2 September 2019) from a list of genes (2000 ± 2-fold, *p* < 0.05). (**B**) *WNT7B*, *WNT9A* and *WNT10B* expression from the RNA-seq analysis of doxycycline-induced LNCaP/Luc/PHB^cDNA^ cells. (**C**) qPCR validation of WNT gene downregulation in LNCaP/Luc/PHB^cDNA^ cells with doxycycline treatment for 48 h. (**D**) q-PCR validation of PHB and *WNT7B*, *WNT9A* and *WNT10B* expression in doxycycline-treated LNCaP/Luc/PHB^siRNA^ cells after 48 h. Data were normalised to housekeeping genes *RPL19*, *GAPDH* and *ACTB*. * *p* ≤ 0.05, ** *p* ≤ 0.01.

**Figure 4 ijms-24-09919-f004:**
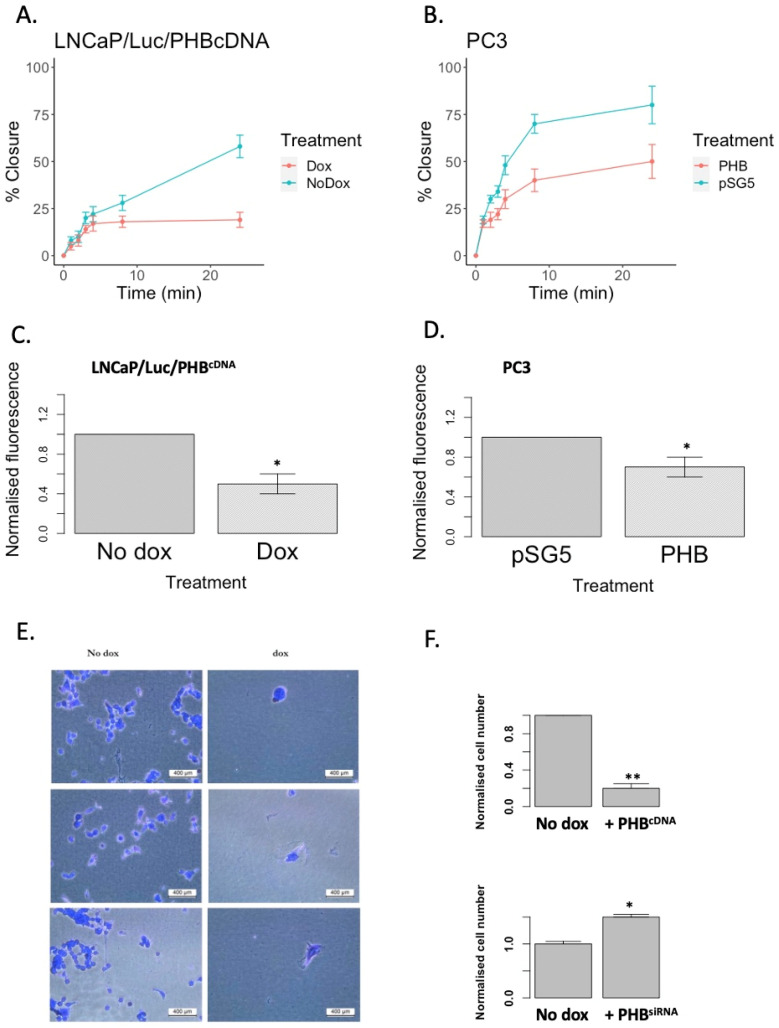
*PHB* overexpression inhibits the migration of both androgen-dependent and androgen-independent PC cells. (**A**,**B**) Graphs indicating the % closure of a scratch (wound-healing) assay of LNCaP/Luc/PHB^cDNA^ cells with or without doxycycline (left-hand side) and PC3 cells (right-hand side) transfected with pSG5-PHB (or empty vector) over 21–48 h. (**C**,**D**) Bar graphs indicating the number of cells invading through a Matrigel layer and across a membrane. Left-hand side shows LNCaP/Luc/PHB^cDNA^ cells with or without doxycycline and right-hand side shows PC3 cells transfected with pSG5-PHB (or empty vector). (**E**) Adhesion assay for LNCaP/Luc/PHBcDNA cells. Cells were stained with crystal violet after adhering to Matrigel basement membrane. Scale bar = 400 μm. (**F**) Bar graphs representing cell number counts from cellular adhesions assays—upper graph LNCaP/Luc/PHB^cDNA^ and lower graph LNCaP/Luc/PHB^siRNA^ cells. * *p* ≤ 0.05, ** *p* ≤ 0.01.

**Figure 5 ijms-24-09919-f005:**
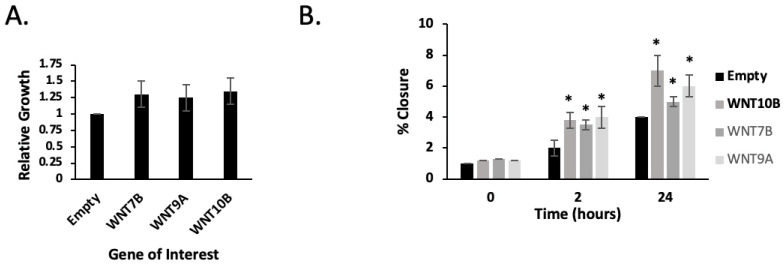
Overexpression of *WNT7B*, *WNT9A* and *WNT10B* increase LNCaP cell growth and motility in vitro. (**A**) Bar graph showing relative growth of LNCaP cells transiently transfected with either empty plasmid or expression plasmids for *WNT7B*, *WNT9A* and *WNT10B*, as determined by MTT assay. (**B**) Graphs indicating the % closure of a scratch (wound-healing) assay of LNCaP cells transiently transfected with either empty plasmid or expression plasmids for *WNT7B*, *WNT9A* and *WNT10B* for 0–24 h. * *p* ≤ 0.05.

**Figure 6 ijms-24-09919-f006:**
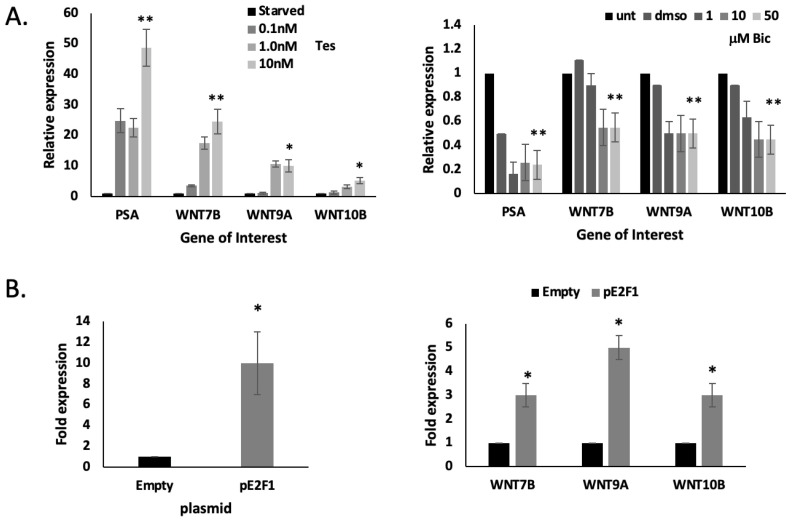
*WNT7B*, *WNT9B* and *WNT10B* are AR-, PHB- and E2F1-responsive. (**A**) Q-PCR analysis of *WNT7B*, *WNT9* and *WNT10B* (and PSA (KLK3) control) from LNCaP cells (48 h). Left-hand side shows fold change in gene expression from LNCaP cells grown in charcoal-stripped serum that was then treated with 0–10 nM testosterone for 24 h. Right-hand side shows LNCaP cells treated with bicalutamide (0–50 μM) for 24 h. (**B**) Q-PCR analysis of *WNT7B*, *WNT9B* and *WNT10B* and *E2F1* expression from LNCaP cells transfected with p-E2F1 plasmid (or empty vector). Gene expression given as fold change and normalised to *GAPDH*, *RPL19* and *ACTB*. * *p* ≤ 0.05, ** *p* ≤ 0.01.

**Figure 7 ijms-24-09919-f007:**
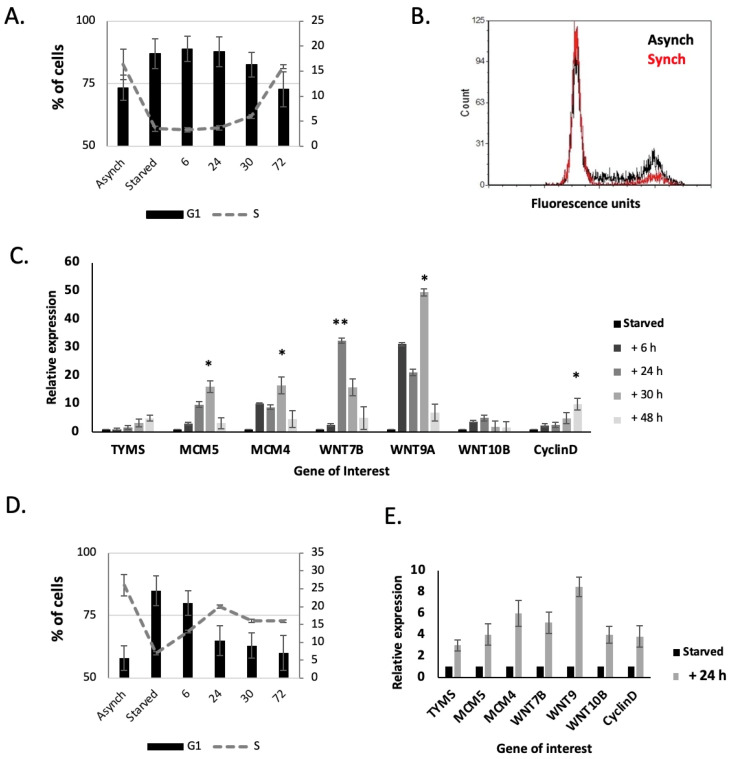
*WNT7B*, *WNT9A* and *WNT10B* expression are linked to the cell cycle. (**A**) Flow cytometry analysis showing the G1 and S phase % of LNCaP cells synchronised in isoleucine-free media and then released into normal media for 72 h. (**B**) Cell cycle distribution of LNCaP cells either asynchronous or synchronised for 72 h. (**C**) Q-PCR analysis of *WNT7B*, *WNT9A* and *WNT10B* (and cell-cycle-related genes) from LNCaP cells released from isoleucine synchronisation. (**D**) FACS analysis showing the G1 and S phase % of PC3 cells synchronised in isoleucine-free media and then released into normal media for 72 h. (**E**) Q-PCR analysis of *WNT7B*, *WNT9A* and *WNT10B* (and cell-cycle-related genes) from PC3 cells released from isoleucine synchronisation. Normalised to *GAPDH*, *RPL19* and *ACTB*. * *p* ≤ 0.05, ** *p* ≤ 0.01.

## Data Availability

Data supplied as Appendix A.

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
