# Peer review of "Prohibitin Links Cell Cycle, Motility and Invasion in Prostate Cancer Cells"

_ijms, 2023, doi:10.3390/ijms24129919_

Round 1
Reviewer 1 Report
Koushyar et al, have studied the role of PHB as a tumor suppressor in prostate cancer as a follow-up paper to their previous work. Experimentally PHB is shown here to mediate migration and adhesion of Pca cells, and metastasis in vivo. Previous RNA seq data suggest that Wnts, as well as integrin-mediated cell adhesion are regulated by PHB. Here connection between Wnts and PHB in studied experimentally and suggested to connect PHB to regulation of metastasis.
Manuscript is not easy to understand, and the figures do not form a logical story together. Abstract should be rewritten, now it is not clear what is background information and what are the experiments carried out now and results of this study. In addition, the results section should be rewritten due to the same reason. For example, 83-85 and 173-175 it is not clear what was done previously and what is done in this study as a new experiment. Many parts of the text need to be clarified, e.g. 65-68: sentence not clear, 83-85: remove repetition in text, 183-186: the connection between these sentences is not clear, this chapter should be clarified.
Fig 1. Several different prostate cancer datasets are available. Please confirm in multiple datasets of PC that PHB is downregulated in metastasis, and it is important to show if this has prognostic significance. Does PHB expression correlate with EMT, cell adhesion and metastasis-related genes in general in patient cohorts? More extensive in silico work could make the paper stronger.
Fig 2. Previously done RNA seq data and PHB knockdown experiments done here suggest that PHB regulates Wnts. In Fig 2 the authors show in silico that wnts are overexpressed in metastatic pca when compared to normal prostate, but this shows no connections to PHB. It should be analyzed if PHB expression correlates with Wnts in silico in different patient cohorts.
Fig 3. The authors should show experimentally that PHB-mediated cell migration and adhesion/metastasis is regulated via Wnts. In fig 3E, how is it known that adhesion is regulated or is the result due to that the cell number reduced? Is there more Wnts expression after PHB knockdown in vivo (Fig 1)?
Resolution of figures is not good and should be improved. Text in figures is often too small font (e.g. 2C). Layout of the manuscript is not good, it should be improved so that figure texts are directly under the figures. Fig 4 is over the margins.
Author Response
Comments and Suggestions for Authors
Koushyar et al, have studied the role of PHB as a tumor suppressor in prostate cancer as a follow-up paper to their previous work. Experimentally PHB is shown here to mediate migration and adhesion of Pca cells, and metastasis in vivo. Previous RNA seq data suggest that Wnts, as well as integrin-mediated cell adhesion are regulated by PHB. Here connection between Wnts and PHB in studied experimentally and suggested to connect PHB to regulation of metastasis.
Manuscript is not easy to understand, and the figures do not form a logical story together. Abstract should be rewritten, now it is not clear what is background information and what are the experiments carried out now and results of this study. In addition, the results section should be rewritten due to the same reason. For example, 83-85 and 173-175 it is not clear what was done previously and what is done in this study as a new experiment. Many parts of the text need to be clarified, e.g. 65-68: sentence not clear, 83-85: remove repetition in text, 183-186: the connection between these sentences is not clear, this chapter should be clarified.
We have extensively re-written the manuscript to take on the reviewers comments and to try and clarify the work done previously and what is new to the manuscript. This has been done in the abstract, introduction and results sections.
Fig 1. Several different prostate cancer datasets are available. Please confirm in multiple datasets of PC that PHB is downregulated in metastasis, and it is important to show if this has prognostic significance. Does PHB expression correlate with EMT, cell adhesion and metastasis-related genes in general in patient cohorts? More extensive in silico work could make the paper stronger.
Fig 2. Previously done RNA seq data and PHB knockdown experiments done here suggest that PHB regulates Wnts. In Fig 2 the authors show in silico that wnts are overexpressed in metastatic pca when compared to normal prostate, but this shows no connections to PHB. It should be analyzed if PHB expression correlates with Wnts in silico in different patient cohorts.
We have added more information to the datasets studied – these have now been added to more extensive figure 2. This should clearly show PHB and WNTs in metastasis and regression analysis done on these genes. Supplemental figure S2 shows the disease free survival data for PHB and the WNT family studied.
Fig 3. The authors should show experimentally that PHB-mediated cell migration and adhesion/metastasis is regulated via Wnts. In fig 3E, how is it known that adhesion is regulated or is the result due to that the cell number reduced? Is there more Wnts expression after PHB knockdown in vivo (Fig 1)?
We have added a new figure (Fig 5) showing that cell migration is enhanced with Wnt7b, 9A and 10B overexpression.
Resolution of figures is not good and should be improved. Text in figures is often too small font (e.g. 2C). Layout of the manuscript is not good, it should be improved so that figure texts are directly under the figures. Fig 4 is over the margins.
We have redrawn several figures and have improved the size of the fonts, as well as reformatted the sizes of the figures.
Reviewer 2 Report
The study is well done, the material is large enough and the methods look reliable. However the study is based on extensive and very recent literature, gives some new information and this warrants its publication.
Author Response
Comments and Suggestions for Authors
The study is well done, the material is large enough and the methods look reliable. However the study is based on extensive and very recent literature, gives some new information and this warrants its publication.
We thank the reviewer for their comments.
Reviewer 3 Report
The manuscript submitted by Koushyar investigated the role of prohibitin on cell cycle and motility. Most experiments are performed with the established prostate cancer cell line LNCaP after genetic up- and down-regulation of PHB. They observed that Wnt genes, the cell cycle and the migratory behavior of the cells were affected. This molecular pathway may be a putative therapeutic target and/or prognostic marker for patients with advanced stage of PC. However, most of the presented data are corealtive and would need additional validation in another cell model. Moreover, I was wondering about the specificity of the observed connection between PHB-AR-Wnt. Currently the data illustrate only correlative connections. Direct connections e.g. with CHIPSeq or IP would increase the impact of the findings. As well as co-expression to localize the key markers with immunfluorescence analysis. Additionally, the sensitivity of PHB-cDNA cells to androgen-deprivation or chemotherapy would be interesting to adress.
Beside this major concerns, I have some minor remarks:
Introduction: Is there a feedback loop between AR and PHB? How physiologic are the conditions with the enforced overexpression or genetic inhibition?
Results:
Section 2.1: what is known about the cross-reactivity of murine androgen with human AR and how does this affect the results obtained with xenograft models? Moreover, I was wondering about the found liver metastasis. Why metastatic spread into liver? Unusal origin for PCa. Did you validated the human origin, e.g. genetically additionally?
Results Section 2.2: What is known about the correlation of PHB gene and protein expression? What is the half-life of prohibitin?
Figure 1A: Please add full description of the Y-axis and show individual measurements. I would suggest to include the tumor growth curves and/or tumor uptake curves (Kaplan-Meier).
Figure 1C: Add liver as origin in the graphics and image quantification. the histological section seems to have different structures. Also in Dox- sample are positive nuclei seen. Why?
Figure 1C and 1D: I would suggest to add Prohibitin immunohistochemistry or at least PHB gene expression of the xenograft tumors and metastasis. Do you have Ki67 IHC data to illustrate proliferative activity?
Figure 1F: The Y-axis is cut. Please unify with Figure 1E. Add numbers below, not into the box blot. Add individual measurements. Please add into the legend how many patients are included per group.
Results section 2.3: Do the identified wnt7b and 10b belonging to the canonical or non-canonical Wnt pathway? It would be of interest to show if the siRNA-PHB can rescue the PHB-cDNA phenotype and vice versa.
Results section 2.4: The connection of cell cycle and cell adhesion with PHB is not fully clear. Here, the authors applied another cell line. I was wondering about the PHB gene and protein expression level of PC3 in comparision to LNCaP and why different plasmid were used?
Figure 2B: Please add PHB and AR as control. Compare siRNA vs. cDNA line.
Figure 2C: Legend is too small to read.
Figure 2D: Add correlation analysis of PHB, AR and Wnts. Are the survival data available to add Kaplan-Meier curves?
Figure 3SB: Can the Wnt7B, 9A and 10B transfection into LNCAP-siPHB rescue of PHB effect?
Results section 2.6: what is the cellular localization - cytoplasmic vs. nuclear - after PHB overexpression and knock-down? Moreover, I was wondering how the genetic PHB alteration affect cell doubling time and would suggest to add proliferation curves. Additionally, please comment on hy HeLa and not PC3 cells were used for the synchronization experiments and how the PHB protein and gene expression is in HeLa.
Figure 3A, 3B, 3C, 3D, 3F: statistic is missing
Figure 3E: legend is too small
Figure 4: statistic is missing in all
Figure 5A: the appreviation FACS=fluorescence activated cell sorting, but no cells are purified. Here, the flow cytometry method was applied. Please change this
Figure 5B: why do you see cells in G2 phase in the synchronized sample?
Figure 5C, 5F: please add PHB and AR gene expression dynamics as control.
Which results would you expect with the RWPE-1 cell line as 'normal' or benign control after PHB up- and down-modulation? Maybe it would be of interest to add this control.
Discussion: I am still not confident, if the observed connections are direct or indirect; specific or unspecific. Please discuss it more critically.
Method section: why are different synchronization methods used for LNCaP and HeLa? How comparable are the obtained results? Moreover, I do not understand the adhesion assay. There is no washing step after cell plating and before fixation? How specific is cell adhesion to matrigel, is this an active process?
Author Response
Comments and Suggestions for Authors
The manuscript submitted by Koushyar investigated the role of prohibitin on cell cycle and motility. Most experiments are performed with the established prostate cancer cell line LNCaP after genetic up- and down-regulation of PHB. They observed that Wnt genes, the cell cycle and the migratory behavior of the cells were affected. This molecular pathway may be a putative therapeutic target and/or prognostic marker for patients with advanced stage of PC. However, most of the presented data are corealtive and would need additional validation in another cell model. Moreover, I was wondering about the specificity of the observed connection between PHB-AR-Wnt. Currently the data illustrate only correlative connections. Direct connections e.g. with CHIPSeq or IP would increase the impact of the findings. As well as co-expression to localize the key markers with immunfluorescence analysis. Additionally, the sensitivity of PHB-cDNA cells to androgen-deprivation or chemotherapy would be interesting to adress.
We have repeated and included more experiments done with the other prostate cancer cell line PC3, and carried out regression analysis in 2 online data sets. We have previously published data showing PHB and E2F coIP, and we would not expect PHB and Wnt to coIP, just that Wnts would be regulated by PHB. PHB does not bind DNA in itself, and just inhibits E2F activity therefore chip-seq would be inconclusive. We agree that sensitivity of cells to androgen deprivation would be interesting, this is outside the scope of the current paper, and has been partially addressed in our previous papers.
Beside this major concerns, I have some minor remarks:
Introduction: Is there a feedback loop between AR and PHB? How physiologic are the conditions with the enforced overexpression or genetic inhibition?
In the clinical data sets analysis for PHB expression, we have observed a maximum of x4 phb expression between metastasis and normal prostate tissues (presented in figure 2). Our western blots here are in the region of x5-x7 PHB expression and therefore they are only slightly higher than physiological.
Results:
Section 2.1: what is known about the cross-reactivity of murine androgen with human AR and how does this affect the results obtained with xenograft models? Moreover, I was wondering about the found liver metastasis. Why metastatic spread into liver? Unusal origin for PCa. Did you validated the human origin, e.g. genetically additionally?
No crossreactivity has been documented with the antibody used. Additionally, the histological morphology is quite different in the liver metastasis plus there is the presence of luciferase activity which all indicate that these are LNCaP deposits. Agreed - the liver is an unusual location for prostate cancer spread, but liver metastasis are present in the GEO clinical datasets used, therefore are not unheard of. Also metastatic migration may be different from a xenograft grown on the flank compared to spread from primary organs in humans.
Results Section 2.2: What is known about the correlation of PHB gene and protein expression? What is the half-life of prohibitin?
We cannot determine the half life of PHB and this is beyond the scope of this article.
Figure 1A: Please add full description of the Y-axis and show individual measurements. I would suggest to include the tumor growth curves and/or tumor uptake curves (Kaplan-Meier).
The Y axis shows the relative tumour volume (RTV), which is a standard from of tumour volume measurement in mouse xenograft experiments. The tumour growth curve data has been previously published by our group, therefore we are keeping this data to a minimum to avoid replication.
Figure 1C: Add liver as origin in the graphics and image quantification. the histological section seems to have different structures. Also in Dox- sample are positive nuclei seen. Why?
Figure is clearly labelled as liver sections. The histological sections will have a different structure due to the presence of LNCaP metastatic deposits, these are disrupting the normal liver structure.
Figure 1C and 1D: I would suggest to add Prohibitin immunohistochemistry or at least PHB gene expression of the xenograft tumors and metastasis. Do you have Ki67 IHC data to illustrate proliferative activity?
Again these have been extensively published in our previous work. PHB immunohistochemistry and the replication marker phospho-histone H3 (rather than Ki67) have been published from these tumours.
Figure 1F: The Y-axis is cut. Please unify with Figure 1E. Add numbers below, not into the box blot. Add individual measurements. Please add into the legend how many patients are included per group.
We have redrawn these figures.
Results section 2.3: Do the identified wnt7b and 10b belonging to the canonical or non-canonical Wnt pathway? It would be of interest to show if the siRNA-PHB can rescue the PHB-cDNA phenotype and vice versa.
We have added a section in the discussion to address the wnt pathways.
Results section 2.4: The connection of cell cycle and cell adhesion with PHB is not fully clear. Here, the authors applied another cell line. I was wondering about the PHB gene and protein expression level of PC3 in comparision to LNCaP and why different plasmid were used?
PC3 cells express lower levels of PHB (previously published by us and others), which also supports the role of lowered PHB in a more metastatic cell line. Different plasmids were used as PC3 cells could not be made with the doxycycline system due to inherent resistance to selection agents. The plasmids resulted in similar levels of PHB protein expression.
Figure 2B: Please add PHB and AR as control. Compare siRNA vs. cDNA line.
Figure 2C: Legend is too small to read.
Figure 2D: Add correlation analysis of PHB, AR and Wnts. Are the survival data available to add Kaplan-Meier curves?
These have been redrawn and Kaplan meier curves added to supplemental.
Figure 3SB: Can the Wnt7B, 9A and 10B transfection into LNCAP-siPHB rescue of PHB effect?
We have not tried this but we would not expect this to rescue the phenotype. Our hypothesis is that PHB inhibits E2F, which is required for cell cycle entry. Wnt expression here is related to cell cycle entry, but we would not expect wnt overexpression to affect cell cycle.
Results section 2.6: what is the cellular localization - cytoplasmic vs. nuclear - after PHB overexpression and knock-down? Moreover, I was wondering how the genetic PHB alteration affect cell doubling time and would suggest to add proliferation curves. Additionally, please comment on hy HeLa and not PC3 cells were used for the synchronization experiments and how the PHB protein and gene expression is in HeLa.
We have added PC3 cell data to the main paper, and moved the HeLa data to the supplemental.
Figure 3A, 3B, 3C, 3D, 3F: statistic is missing
Figure 3E: legend is too small
Figure 4: statistic is missing in all
We have added stats to all the relevant figures.
Figure 5A: the appreviation FACS=fluorescence activated cell sorting, but no cells are purified. Here, the flow cytometry method was applied. Please change this
We have amended this.
Figure 5B: why do you see cells in G2 phase in the synchronized sample?
The synchronisation will not be 100% perfect and there will always be some residual G2/M cells with any chemical synchronisation method. The main result is the much higher % in G1.
Figure 5C, 5F: please add PHB and AR gene expression dynamics as control.
The PHB and AR gene dynamics are not substantially changed in these experiments, and are therefore not included.
Which results would you expect with the RWPE-1 cell line as 'normal' or benign control after PHB up- and down-modulation? Maybe it would be of interest to add this control.
In previous analysis using this cell line, we have noticed that PHB expression did not follow a pattern that was observed for other prostate cell lines according to their described ‘aggressiveness’, and was a distinct outlier when comparing LNCaP, VCap, DuCaP, PC3, C42 and therefore would be a confounding result. This was due to housekeeping genes both RNA and protein being very different in this cell line, making comparisons very difficult.
Discussion: I am still not confident, if the observed connections are direct or indirect; specific or unspecific. Please discuss it more critically.
We have rewritten aspects of the discussion.
Method section: why are different synchronization methods used for LNCaP and HeLa? How comparable are the obtained results? Moreover, I do not understand the adhesion assay. There is no washing step after cell plating and before fixation? How specific is cell adhesion to matrigel, is this an active process?
We have now used LNCaP and PC3 cell studies for comparison in the main paper, and have moved the HeLa cells data to the supplemental. We used this much more harsh method of synchronisation with HeLa cells as it is very well described and tolerated by the cells, and results in a very tight synchronisation. Synchronisation of prostate cancer cells is less well described in mainstream journals. This method was not tolerated by the prostate cancer cells. HeLa cell data no longer forms part of the main paper.